# Changes in the HIV Epidemic in Lower Silesia, Poland, Between 2010 and 2020: The Characteristics of the Key Populations

**DOI:** 10.3390/v16091445

**Published:** 2024-09-11

**Authors:** Aleksandra Kozieł, Aleksandra Cieślik, Łucja Janek, Aleksandra Szymczak, Igor Domański, Brygida Knysz, Bartosz Szetela

**Affiliations:** 1Department of Infectious Diseases, Liver Disease and Acquired Immune Deficiencies, Wroclaw Medical University, 50-367 Wroclaw, Polandaleksandra.szymczak@umw.edu.pl (A.S.); igor.domanski@student.umw.edu.pl (I.D.); brygida.knysz@umw.edu.pl (B.K.); bartosz.szetela@umw.edu.pl (B.S.); 2Statistical Analysis Centre, Wroclaw Medical University, 50-367 Wroclaw, Poland; 3All Saint’s Clinic, Wrocławskie Centrum Zdrowia SP ZOZ, 50-136 Wrocław, Poland

**Keywords:** HIV epidemic, Poland, key populations

## Abstract

The HIV (Human Immunodeficiency Virus) epidemic remains a significant public health issue, requiring ongoing access to preventive methods. This study aimed to analyze the evolution of the HIV epidemic in Lower Silesia from 2010 to 2020, focusing on the key populations. A retrospective analysis of the medical records from newly diagnosed HIV patients at a major HIV clinic in Wroclaw was conducted, examining demographic data, infection routes, and laboratory results. An 84% increase in newly diagnosed HIV cases was observed over the decade, with the most common route of infection being sex between men (70% among those with a known infection route). These patients were generally in better clinical condition compared to their heterosexual counterparts, as indicated by a higher median CD4+ T cell count (465/μL vs. 250/μL). The changes in clinical status and infection routes were statistically significant. The HIV epidemic in Lower Silesia has shifted, with a notable rise in new infections among men who have sex with men. Heterosexual patients were often diagnosed at more advanced stages. Prevention strategies should adapt to these changing trends, with education and testing accessibility remaining priorities nationwide.

## 1. Introduction

Human immunodeficiency virus (HIV) remains a major global public health problem. In total, 38.4 million people were estimated to be living with HIV worldwide at the end of 2021, with two-thirds (25.6 million) in the WHO’s African region. By that date, 40.1 million deaths were reported to be the result of an HIV infection [1].

In Poland, epidemiological records have been monitored by the National Institute of Public Health—National Institute of Hygiene since 1985. Until the end of 2021, an HIV infection has been diagnosed in 27,557 individuals, of which one in two does not know that they can spread the virus further [2]. These data should be interpreted with great caution, because there is no thorough epidemiological surveillance in Poland; no exclusion of results performed on the same patient; and migration and deaths are not reliably taken into account. Some doubts remain about the correctness of these figures.

At the beginning of the HIV epidemic in 1985, the predominant route of an HIV infection was intravenous drug use (IDU). Between 1999 and 2004, infections from IDU accounted for 78.6% of all cases [3]. In the following years, the proportion of infections among men who have sex with men (MSM) gradually increased. In 2009, MSM accounted for 34.2% of new diagnoses, and in 2010 they accounted for 51.9% [4]. This trend has also been observed in other countries of the European Union [5].

For years, Lower Silesia (a voivodeship in Poland) had one of the worst epidemiological situations in Poland. Between 1999 and 2004, the incidence of HIV in the area was 34.7/1,000,000 inhabitants and was the highest in the country [3]. Currently, this region also presents one of the highest incidence rates in the country (the number of patients diagnosed with infection residing in a given area in a given year/100,000 inhabitants). It reached 61.6 in 2021 [6].

The aim of this study was to describe the changes in the epidemiological situation in Lower Silesia between 2010 and 2020 and to describe the key populations.

## 2. Materials and Methods

A retrospective analysis was performed based on the data from medical records from an HIV clinic in Wroclaw. The years 2010, 2016, and 2020 were chosen for analysis to compare the patients’ clinical status at diagnosis over a decade at similar time intervals (Table 1).

The following data were collected during the first visit: the demographic data, route of infection, initial viral load, and CD4+ T cell count; whether the patient remembered symptoms that could be consistent with acute retroviral disease; and whether he had symptoms of AIDS, as well as coexisting infections including HBV (defined as HBsAg-positive), HCV (defined as anti-HCV-positive), and syphilis (defined as a positive VDRL test). 

The route of HIV infection was determined based on medical history. Each patient was assigned one of three routes of infection: HTXs (heterosexuals), MSM (men who have sex with men), and IDUs (intravenous drug users). The patients in whom only one route of infection could be attributed were eligible for the analysis.

The assumption of a normality of distribution was checked with the Student’s t-test and the ANOVA test. The assumption of the normality of distribution was not met. The Mann–Whitney U test was used to determine the presence of anti-HCV antibodies (in two groups) depending on age (the quantitative variable) for each year. The Kruskal–Wallis test was used to examine the relationship of the quantitative variables: age, viral load, the number of the CD4+ T cell count (a quantitative variable) over time (in three groups), and the number of the CD4+ T cell count between the routes of infection (in three groups) for each year. A Dunn’s post hoc test was used to examine between which routes of infection there were statistically significant differences in the number of the CD4+ T cell count for each year.

The assumption of the number of observations (*n* < 5 in ≤20% of cells) was checked for the chi-squared test. The Pearson chi-squared test of independence without the Yates correction for continuity was used to examine the effect of year on the proportions of infection routes and on the occurrence of anti-HCV antibodies. The Fisher–Freeman–Halton test was used to examine the influence of the route of infection on the occurrence of anti-HCV antibodies for each year. 

For the characteristics of the study group and for post hoc purposes, the significance level of α = 0.05 was adopted. A multiple comparisons Bonferroni correction was applied to the α-values obtained from the tests.

The Fisher–Freeman–Halton test was performed in R Studio (version 4.3.1) using the packages: library (“stats”) [7] and library (“openxlsx”) [8]. The remaining tests and post hoc analysis were performed in Statistica 13.3.721.1.

## 3. Results

A total of 202 patients (27 women, including 22.22% pregnant, and 175 men) were included in the analysis. In total, 43 patients (nine women and 34 men) were analyzed in 2010; 80 patients (seven women and 73 men) in 2016; and 79 patients (eleven women and 68 men) in 2020. In total, 128 of them acquired HIV through male-to-male sexual contact, 34 through heterosexual contact, 20 through intravenous drug use, and for 20 patients the route of infection remained unknown. Acute retroviral disease was diagnosed in 52 patients and AIDS in 39. In the analyzed group, twenty-six patients were infected with HCV and four with HBV. Positive VDRL results were seen in 31 patients. Among the entire group of women, six were diagnosed with HIV during pregnancy (Table 1). 

### 3.1. Characteristics of Patients at the First Visit 

There was no statistical difference in the patients’ age in the three time periods (Kruskal–Wallis test: H = 0.88, df = 2, and *p* = 0.644). The average median age of the patient at diagnosis was about 32 years. The viral load (Kruskal–Wallis test: H = 1.77, df = 2, and *p* = 0.414) and the number of the CD4+ T cell count also did not differ in the time periods (Kruskal–Wallis test: H = 1.94, df = 2, and *p* = 0.397). The average median number of the viral load was 28717 copies/mL and the CD4+ T cell count was 395 cells/ μL (Table 2). 

### 3.2. Route of Infection

The most common route of infection was MSM—70.33% (among patients with a known route of infection). In 2010, the second most common route of infection was among IDUs and, in subsequent years, HTXs (Pearson’s chi-squared test: χ^2^ = 30.94, df = 4, and *p* < 0.001) (Table 3).

### 3.3. CD4+ and the Route of Infection

The CD4+ T cell count was statistically significant dependent on the route of infection in 2010 and 2016 (Kruskal–Wallis tests: H = 8.46, df = 2, and *p* = 0.015 and H = 17.66, df = 2, and *p* < 0.001.). Each year, MSM had the highest number of the CD4+ T cell count. In 2010, IDUs had the second highest number of the CD4+ T cell count, followed by HTXs. In 2016, we saw a change—the CD4+ T cell count among HTXs was higher than among IDUs. In 2020, the differences in the number of the CD4+ T cell count dependent on the route of infection were statistically insignificant (Kruskal–Wallis test: H = 5.94, df = 2, and *p* = 0.051) (Table 4) (Figure 1). 

In total, 28.81% of MSM, 67.65% of HTXs, and 60% of IDUs were diagnosed as late presenters (LPs), indicating individuals diagnosed with either a CD4+ T cell count below 350 cells/μL or diagnosed with an opportunistic infection, regardless of the CD4+ T cell count at the time of diagnosis [9].

### 3.4. Hepatitis C Infection in the Three Periods

The prevalence of positive anti-HCV results at the first visit in 2010, 2016, and 2020 was 40%, 13.16%, and 3.45%, respectively (Figure 2). The prevalence of anti-HCV antibodies differs significantly in the three analyzed years (chi-squared test: χ^2^ = 22.93, df = 2, and *p* < 0.001) and is significantly dependent on the route of infection in 2010 (Fisher’s test: *p* < 0.001). IDUs stand out (75%) in relation to the others—HTXs (17%) and MSM (8%). For the remaining years, no statistical significance was demonstrated for different routes of transmission (Fisher’s test: *p* = 0.064 and *p* > 0.999). 

No statistical significance was found between the presence of anti-HCV antibodies and gender in any of the years (2010—Fisher’s test: *p* > 0.999; 2016—Pearson chi-squared test: χ^2^ = 10.28, df = 1, and *p* = 0.001; and 2020—Fisher’s test: *p* > 0.999)

The presence of anti-HCV antibodies was statistically significant dependent on age only in 2016 (Mann–Whitney U test: Z = −1.39 and *p* = 0.001) and statistically insignificant in 2010 (Student’s t-test: *p* = 0.201, t = 1.30, and df = 32) and 2020 (Mann–Whitney U test: Z = 1.51 and *p* = 0.150). 

### 3.5. Syphilis 

The prevalence of positive VDRL results in 2010, 2016, and 2020 was 16% (*N* = 7), 12.5% (*N* = 10), and 17.7% (*N* = 14), respectively. There was no statistical significance (Pearson chi-squared test: χ^2^ = 2.47, df = 2, and *p* = 0.291).

## 4. Discussion

According to data from the National Institute of Public Health—National Institute of Hygiene, Poland experienced a 33.8% increase in new HIV infections between 2010 and 2019. However, the emergence of the COVID-19 pandemic in 2020 significantly disrupted this trajectory [10]. 

However, the number of new diagnoses in the analyzed HIV clinics almost doubled between 2010 and 2020. This is probably due to the increased awareness among residents of Wroclaw and the Lower Silesian voivodeship, as well as the large number of HIV tests performed compared to other regions in Poland [11,12,13]. In contrast, the trends in reported HIV diagnoses across Europe have been on a downward trajectory since 2012. During this period, the regularly reported HIV diagnosis rate in EU/EEA countries decreased from 6.4 cases per 100,000 inhabitants to 3.7 cases in 2021 [14]. These statistics underscore a worrying reality: in Poland, the HIV epidemic appears to remain unchecked when juxtaposed with the overall trend observed across Europe. 

The median age of patients has changed insignificantly over the years in the study group. These data do not coincide with the National Health Fund (NHD) report from 2022, which states that, over recent years, there has been an increase in the age at the time of the diagnosis of an HIV infection [6]. These results suggest to authorities and health care providers that the prevention programs in Lower Silesia should focus on people aged 30 and younger.

In any given year, in a study group, MSM were the most common route of infection. In 2020, MSM accounted for 65% of patients. According to the “HIV/AIDS surveillance in Europe 2022 (2021 data)”, approximately 60% of new cases of HIV infections in Poland in 2021 were among MSM [14]. This figure was similar to the result of this study. The decrease in the number of IDUs among new HIV patients and in the incidence of anti-HCV antibodies is likely due to the implementation of needle exchange, harm reduction, and the methadone substitution program [15]. According to the National Bureau for Drug Addiction’s 2020 report, there were only 20 new cases of HIV infections among IDUs in Poland in 2019, compared to 47 cases in 2010 [16]. In Central Europe, the number of new HIV infections among IDUs decreased from 15% in 2011 to 5% in 2020 [5]. It is worth noting that there has been an opioid substitution treatment program (OST) for heroin users running in the analyzed HIV clinic, so IDUs had fewer barriers in access to testing and treatment. On admittance to this OST center, every patient was tested for HIV, HCV, and syphilis, allowing for rapid treatment. 

Despite the decrease in the number of patients with an HCV co-infection at the time of the HIV diagnosis, an HCV infection was often diagnosed at subsequent visits, especially among MSM (our own observation outside of the scope of this paper, unpublished).

Within the study cohort, across each of the examined years, MSM consistently presented with the least advanced stage of infection, as evidenced by a notably higher median CD4+ T cell count (465/μL), in contrast to IDUs (327/μL) and HTXs (250/μL).

In a study examining the data from 14 out of 17 HIV treatment centers in Poland over the period from 2000 to 2015, notable trends emerged. Specifically, it was also observed that MSM tended to have a higher median CD4+ T cell count at the time of diagnosis (357/μL) compared to other groups—IDUs (302/μL) and HTXs (259/μL) [17].

Similarly, a larger scale analysis covering 21 European countries and spanning 2010 to 2018 found that MSM tended to have a higher median CD4+ T cell count at diagnosis (~440/μL) compared to heterosexual individuals (~300/μL) [18]. This suggests that MSM may be getting diagnosed earlier, possibly due to greater awareness and more frequent testing among this group.

Recent large-scale research has also highlighted the trend of men who have sex with men (MSM) having a higher CD4+ T cell count at diagnosis compared to heterosexual individuals (HTXs). This has led to new theories suggesting that more aggressive strains of HIV-1 might be selected due to the apparent difficulty in infecting HTX individuals [19]. 

It is crucial to highlight the significantly low CD4 + T cell count observed among HTXs in our study. Nearly 70% of HTXs were classified as "late presenters" (LPs), indicating individuals diagnosed with either a CD4+ T cell count below 350 cells/μL or diagnosed with an opportunistic infection, regardless of the CD4+ T cell count at the time of diagnosis [9].

Supporting evidence from a study spanning 2009 to 2016, conducted at a smaller Voluntary Counseling and Testing (VCT) facility in Poland, revealed a similar trend, with almost 69% of HTX subjects categorized as late presenters [20]. Furthermore, another study, encompassing data from 13 out of 17 Polish clinics during 2016 and 2017, substantiated this finding, demonstrating that 56% of HTXs were late presenters [21]. Notably, both studies identified older individuals, injecting drug users (IDUs), and heterosexual individuals as more prone to late presentation compared to men who have sex with men (MSM). 

A study from Scotland that collected data from 1999 to 2003 found the same conclusions. Of the 78 HIV-infected heterosexual individuals diagnosed, 45% were late presenters. Fewer homosexual men were late presenters [22].

An American study analyzing data from 2000 to 2009 also showed that MSM have a lower risk of late diagnosis compared to other groups—27% of MSM were diagnosed late, compared to 47% of other groups [23].

A statistical analysis of the VCT facility’s data in Poland from 2018 further underscores the heightened risk of late presentation among heterosexual individuals. This elevated risk is likely attributed to the lower perceived susceptibility to the disease among this demographic, resulting in less frequent screening practices [24].

Studies from other European countries also show that HTXs are more likely to be diagnosed late compared to MSM and the percentage of HTXs with a late diagnosis is increasing. 

In Scotland in 2019, 13% of HTXs were diagnosed as LPs compared to 27% in 2023 (vs. 23% among MSM) [25]. 

Based on German statistics, there has been no significant change in the number of late diagnoses or the absolute number of late diagnoses among HTXs in recent years. However, the percentage share of HTXs among late diagnoses is increasing due to a decrease in the number of late diagnoses among MSM year by year [26]. Furthermore, a study conducted in Denmark between 2009 and 2017, which analyzed the data from 1,225 patients, found that the prevalence of LPs was approximately 50.5% and was higher in HTXs (67.3%) than in MSM (34.9%) [27].

No increase in syphilis cases was observed in our study. This may be due to the fact that the sample size was too small, because the data do not coincide with the national data. According to the National Institute of Hygiene, there has been an increase in the number of cases of Treponema pallidum infection since 2006 in Poland [28]. The worldwide situation is similar. In Europe, the number of cases increased by 50% between 2009 and 2018 [29,30]. 

The data from 28 EU/EEA countries from 2021 showed that the majority (77%) of syphilis cases were linked to MSM. From 2012 to 2019, there was a consistent rise in reported syphilis cases among men, predominantly fueled by an increase in instances within the MSM community [31].

## 5. Conclusions

Although new HIV infections among MSM still dominate in Europe, it is important to note the worsening situation among HTX patients. There is an urgent need to strengthen prevention activities for the two key populations of MSM and HTXs. These activities should be tailored to the recipients and their risk factors.

The clinical condition of MSM patients has been improving year by year, while the opposite trend is observed among HTXs, who are commonly being diagnosed at an advanced or very advanced stage. This highlights the divergent directions the HIV epidemic is currently taking.

The training and strengthening of screening and diagnosis among GPs and specialists is needed for the early detection of HIV infections among HTX populations without specific high-risk factors for acquiring an HIV infection. Reducing barriers to testing, both medical and non-medical, including reimbursement for home testing, is necessary. Home testing should also be promoted by doctors.

Access to HIV testing remains problematic in Poland, with VCT facilities only available in large cities. While home testing options now exist, they require greater promotion [32]. The problem also applies to pre-exposure prophylaxis (PrEP). Insufficient knowledge and the difficulty of breaking taboos among family doctors often cause stress for both patients and doctors. Some doctors are unaware of the possibility of prescribing PrEP, and many patients are ashamed to ask for a prescription.

Training GPs to prescribe and identify patients in need of PreP, not only MSM, is also necessary. Continuing and publicizing harm reduction and methadone substitution programs is also important to reduce the number of people infected through IDU.

The continued monitoring of changes is crucial, particularly regarding STIs and their potential impact on HIV transmission risks. Additionally, accurate data collection on new infections and proper reporting to government institutions should be a priority.

## 6. Limitations

The data from the retrospective analysis are incomplete. In addition, the route of the HIV infection could not be determined in twenty patients: six in 2010, seven in 2016, and seven in 2020. The viral load was unknown in ten patients: seven in 2010 and three in 2016. It was difficult to establish a history of acute retroviral disease, as its symptoms are not specific and can be confused by patients with the symptoms of other diseases such as influenza or the common cold [33]. 

The data on the number of new HIV infections in 2020 may also be underestimated due to the COVID-19 pandemic occurring at that time. The epidemic may have contributed to an underestimation of the number of new HIV cases due to the fear of SARS-CoV-2 infection or the inability of patients to reach testing sites.

Incorrect data may relate to the co-occurrence of syphilis with the diagnosis of an HIV infection, as the opposite relationship was observed in Poland in the years in question. This may be due to inadequacies in the retrospective analysis and a delayed diagnosis, as well as too small a study group. It should be taken into account that a positive VDRL test result alone may indicate not only active disease but also a history of syphilis [34].

According to the ‘HIV/AIDS surveillance in Europe: 2022 (2021 data)’ data, Poland has significant deficiencies in supplementing the epidemiological data, which makes it impossible to keep accurate statistics and calculations [5]. In many of the analyses carried out in the aforementioned report, Poland was excluded due to the lack of sufficient statistical data [5].

In conclusion, due to the ongoing changes in the epidemiology of HIV, there is a need for health care and the authorities to adapt. We should change our approach to HIV risk and try to increase the range of information reaching the public. Further reports and studies in this field are needed.

## Figures and Tables

**Figure 1 viruses-16-01445-f001:**
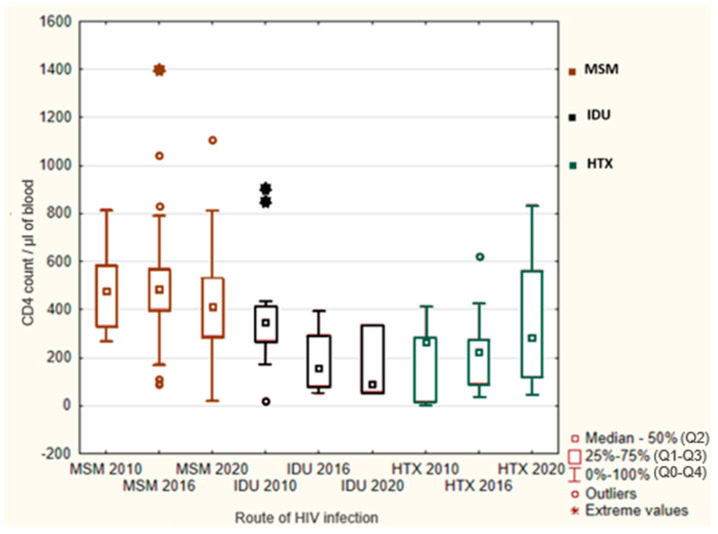
CD4+ T cell count depending on the route of infection in 2010, 2016, and 2020. Outliers—all values less than *Q1* + 1.5 × *IQR* and greater than that, up to *Q3* + 1.5 × *IQR*.

**Figure 2 viruses-16-01445-f002:**
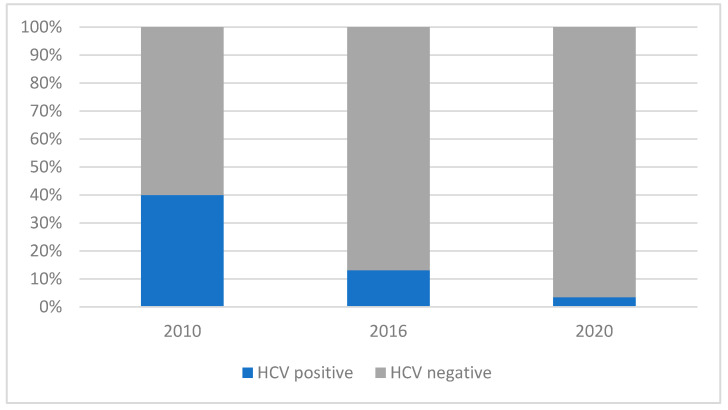
Presence of anti-HCV antibodies dependent on the year of study.

**Table 1 viruses-16-01445-t001:** Characteristics of the study group, *n* (%).

		Year		
Variables	2010	2016	2020	Total
n	43 (21.29)	80 (39.60)	79 (39.11)	202 (100)
Gender:				
Women	9 (20.93)	7 (8.75)	11 (13.92)	27 (13.37)
(Pregnancy)	(1 (11.11))	(0 (0.00))	(5 (45.45))	(6 (22.22))
Men	34 (79.07)	73 (91.25)	68 (86.08)	175 (86.63)
Route of infection:				
MSM	18 (41.86)	59 (73.75)	51 (64.56)	128 (63.36)
HTXs	6 (13.95)	10 (12.50)	18 (22.78)	34 (16.83)
IDUs	13 (30.23)	4 (5.00)	3 (3.79)	20 (9.90)
Unknown	6 (13.95)	7 (8.75)	7 (8.86)	20 (9.90)
Diseases:				
Acute retroviral disease	7 (16.27)	21 (26.25)	24 (30.38)	52 (25.74)
AIDS	6 (13.95)	16 (20.00)	17 (21.52)	39 (19.31)
Anti-HCV-positive	14 (32.56)	10 (12.50)	2 (2.53)	26 (12.87)
HBsAg-positive	0 (0.00)	1 (1.25)	3 (3.80)	4 (1.98)
VDRL-positive	7 (16.27)	10 (12.50)	14 (17.72)	31 (15.35)

*n*—number of observations and route of infection: *MSM*—men who have sex with men, *HTXs*—heterosexuals, *IDUs*—intravenous drug users, and Unknown.

**Table 2 viruses-16-01445-t002:** Comparison of age, viral load, and CD4+ T cell count in 2010, 2016, and 2020, with *n* and *Me (Q1–Q3)*.

		Year	
Variables	2010	2016	2020
Age	43, 33.72	80, 31.35	79, 33.06
	(28.59–38.67)	(26.77–37.16)	(26.97–38.25)
Viral load (copies/mL)	36, 29350.00	77, 27900.00	79, 28900.00
	(7960.00–58600.00)	(15200.00–83000.00)	(8670.00–133500.00)
CD4+ T cell count	43, 359.00	80, 444.50	79, 380.00
	(273.00–481.00)	(246.75–557.25)	(235.50–518.00)

*n*—number of observations and *Me (Q1–Q3)*—median (quartile 1–quartile 3).

**Table 3 viruses-16-01445-t003:** Route of infection and year (only patients with known route of infection included), *n (%)*.

		Year		
Route of Infection	2010	2016	2020	Total
Total	37 (100.00)	73 (100.00)	72 (100.00)	182 (100.00)
MSM	18 (48.65)	59 (80.82)	51 (70.83)	128 (70.33)
HTXs	6 (16.22)	10 (13.70)	18 (25.00)	34 (18.69)
IDUs	13 (35.14)	4 (5.48)	3 (4.17)	20 (10.99)

*n*—number of observations; *MSM*—men who have sex with men; *HTXs*—heterosexuals; and *IDUs*—intravenous drug users.

**Table 4 viruses-16-01445-t004:** The route of infection and CD4+ T cell count depending on the year, with *n* and *Me (Q1–Q3*, ** min–max)*.

		Year		
Route of Infection	2010	2016	2020	Total
MSM	18, 481.0 (337.25–584.75)	59, 488.00 (405.00–571.50)	51, 416.00 (293.00–524.50)	128, 464.50 (330.75–565.50)
HTXs	6, 267.00 * (4.00–414.00)	10, 224.00 (111.75–270.75)	18, 286.50 (127.00–537.25)	34, 250.00 (118.00–413.75)
IDUs	13, 353.00 (274.00–417.00)	4, 157.50 * (48.00–395.00)	3, 93.00 * (60.00–336.00)	20, 327.00 (156.75–386.75)

*n*—number of observations, *Me (Q1–Q3)*—median (quartile 1–quartile 3), *—for subgroups of variables where n is small, minimum–maximum was used, *MSM*—men who have sex with men, *HTXs*—heterosexuals, and *IDUs*—intravenous drug users.

## Data Availability

The data presented in this study are only available on request from the corresponding author due to privacy, legal, or ethical reasons.

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
