# Peer review of "Changes in the HIV Epidemic in Lower Silesia, Poland, Between 2010 and 2020: The Characteristics of the Key Populations"

_viruses, 2024, doi:10.3390/v16091445_

Round 1

Reviewer 1 Report

Comments and Suggestions for Authors

A few of the statistics seem like they could be changed to be more informative.  For example the number of pregnancies was 6 which is 2.97% of all participants, but there were only 27 women.  It seems more informative to note that it was 22% of women, rather than 3% of all participants.

"...

Interestingly, when looking at data from all Voluntary Counseling and Testing (VCT) facilities in Poland from 2018, it was found that only a quarter of patients seeking HIV testing identified as MSM, even though they made up almost half of the male patients. Moreover, a significant portion of men accessing these services (83.9%) reported having had sexual contact with another man at some point in their lives [18].  ..."

Is this saying that men who did not identify as MSM mostly tested negative?  And also that many men who don't identify as MSM have had sex with another man?  Does "male patients" mean HIV-infected?

A few of the discussion points could be moved to the conclusions section of the paper.  Many readers will skip over much of the discussion to read the conclusions, so it can make a difference to place the important points being made in the conclusions section.

In the discussion it is stated "...During this period, the regularly reported HIV diagnosis rate in EU/EEA countries decreased from 6.4 cases per 100,000 inhabitants to 3.7 cases in 2021 [12]. ..." and in the introduction it says the rate in Poland is 61.6 "...Currently, this region also presents one of the highest prevalence in the country (the number of patients diagnosed with infection residing in a given area in a given year/100000 inhabitants). It reached 61.6 in 2021 [6] ..." And surely if most of the infections are among MSM, the incidence rate is very large in that subpopulation.  And I think you meant incidence rate and not prevalence.

Treatment for HIV is not discussed well in the paper.  Especially treatment as prevention, with the idea that people with undetectable viral load are unlikely to transmit virus, should be discussed.  The paper mentions that CD4 count is higher in MSM compared to heterosexuals, but that is at time of diagnosis.

Author Response

Comment 1: A few of the statistics seem like they could be changed to be more informative.  For example the number of pregnancies was 6 which is 2.97% of all participants, but there were only 27 women.  It seems more informative to note that it was 22% of women, rather than 3% of all participants.

Response 1: We agree with this comment. We have changed the percentages of pregnant women, both in the text and in Table 1. 

Comment 2: 

"...

Interestingly, when looking at data from all Voluntary Counseling and Testing (VCT) facilities in Poland from 2018, it was found that only a quarter of patients seeking HIV testing identified as MSM, even though they made up almost half of the male patients. Moreover, a significant portion of men accessing these services (83.9%) reported having had sexual contact with another man at some point in their lives [18].  ..."

Is this saying that men who did not identify as MSM mostly tested negative?  And also that many men who don't identify as MSM have had sex with another man?  Does "male patients" mean HIV-infected?

Response 2: Thank you for that comment. We agree that the above passage is difficult to understand. Furthermore, it does not contribute relevant information to the article. We have decided to remove this paragraph.

Comment 3: A few of the discussion points could be moved to the conclusions section of the paper.  Many readers will skip over much of the discussion to read the conclusions, so it can make a difference to place the important points being made in the conclusions section.

Response 3: We have made the change as suggested. We have moved some sentences to the conclusions. 

Comment 4: In the discussion it is stated "...During this period, the regularly reported HIV diagnosis rate in EU/EEA countries decreased from 6.4 cases per 100,000 inhabitants to 3.7 cases in 2021 [12]. ..." and in the introduction it says the rate in Poland is 61.6 "...Currently, this region also presents one of the highest prevalence in the country (the number of patients diagnosed with infection residing in a given area in a given year/100000 inhabitants). It reached 61.6 in 2021 [6] ..." And surely if most of the infections are among MSM, the incidence rate is very large in that subpopulation.  And I think you meant incidence rate and not prevalence.

Response 5: Thank you for that comment, we agree. We changed the prevalence for incidence rate. 

Comment 6: Treatment for HIV is not discussed well in the paper.  Especially treatment as prevention, with the idea that people with undetectable viral load are unlikely to transmit virus, should be discussed.  The paper mentions that CD4 count is higher in MSM compared to heterosexuals, but that is at time of diagnosis.

Response 6: Dear Reviewer, we do not want to describe the treatment in this paper, as this is a topic for a separate article. Here we wanted to focus only on the epidemiology. 

Reviewer 2 Report

Comments and Suggestions for Authors

In this manuscript, the authors presented an HIV epidemiology study in Lower Silesia, Poland between 2010 and 2020. The authors described the change of the trend of HIV epidemics in this area with high HIV incidence and prevalence. They compared the characteristics of positive cases by surveyed years and transmission route, and studied the presence of viral hepatitis and syphilis coinfection among new diagnosis. I have following comments and suggestions.

1. Can the data from the two clinics well represent the epidemic in the study region?

2. Please reformat Table 1. It is hard to read in the current format.

3. Multiple routes of transmission is possible. For instance MSM and IDU. How did the authors categorize these patients?

4. Figure 1. Suggest using different colors for people with different transmission route.

5. Figure 2 is not very informative. Can the authors generate a figure of HCV coinfection by year and by route of transmission?

Comments on the Quality of English Language

Minor editing of English needed. 

Author Response

Comment 1: Can the data from the two clinics well represent the epidemic in the study region?

Response 1: The data presented concern one clinic, however, the largest in Lower Silesia, and we believe that this group is representative for the area in question. 

Comment 2: Please reformat Table 1. It is hard to read in the current format.

Response 2: Thank you for your comment. We have reworked table 1, in our opinion it should now be more readable.

Comment 3: Multiple routes of transmission is possible. For instance MSM and IDU. How did the authors categorize these patients?

Response 3: Thank you for that comment. Data on the route of infection were established on the basis of the history. Patients with one possible route of infection were taken for analysis. We have added an explanation in the section - materials and methods.

Comment 4: Figure 1. Suggest using different colors for people with different transmission route.

Response 4: We have changed as suggested.

Comment 5: Figure 2 is not very informative. Can the authors generate a figure of HCV coinfection by year and by route of transmission?

Response 5: Dear Reviewer, we do not wish to change the table, as it is in fact statistically significant that the number of HCV co-infections has decreased over the years in question. No significant differences in terms of route of infection were observed for 2016 and 2020. This is probably due to the fact that the study group is too small. We believe that it is more readable to present these data 'in aggregate', without a breakdown by route of infection.